# Polycyclic Aromatic Hydrocarbons in Sediments/Soils of the Rapidly Urbanized Lower Reaches of the River Chaohu, China

**DOI:** 10.3390/ijerph16132302

**Published:** 2019-06-28

**Authors:** Huanling Wu, Binghua Sun, Jinhua Li

**Affiliations:** 1School of Resources and Environmental Engineering, Anhui University, Hefei 230601, Anhui, China; 2Key Laboratory of Aqueous Environment Protection and Pollution Control of Yangtze River in Anhui of Anhui Provincial Education Department, Anqing Normal University, Anqing 246001, Anhui, China; 3School of Life Science, Hefei Normal University, Hefei 230601, Anhui, China

**Keywords:** polycyclic aromatic hydrocarbons, sediment/soil cores, land use types, ecological risk assessment, redundancy analysis

## Abstract

Polycyclic aromatic hydrocarbons (PAHs) are highly teratogenic, persistent carcinogens, and ubiquitous environmental pollutants. To determine the impact of rapid urbanization on sediment/soil PAHs, we collected 30 cm soil cores in ditch wetlands, riverine wetlands, and agricultural lands along the lower reaches of the Shiwuli River feeding Chaohu Lake, China. Ecological risk effects were evaluated by two models based upon Benzo[*a*]pyrene toxic equivalency (TEQ-B*a*P) and total toxic units (TUs). The presence of PAHs, such as BbF, BkF, InP, and BgP, that are known pollutants of concern, suggests certain ecological risks. The concentration of PAHs in the surface layer followed in the order of: ditch wetlands (617.2 ng/g average), riverine wetlands (282.1 ng/g average), agricultural lands (103.7 ng/g average). PAHs in ditch sediments were vertically distributed evenly, and PAHs in agricultural soils were concentrated in the surface soil. In riverine wetland sediments, the 2-, 3-, and 4-ring PAHs had a uniform distribution, whereas the 5- and 6-ring PAHs were concentrated in the surface soil. Redundancy analysis (RDA) explored the correlation between the environmental properties and the occurrence of PAHs. Total organic carbon (*p* = 0.010), percent clay (*p* = 0.020), and distance (*p* = 0.020) were the primary factors in ditch wetlands. Depth (*p* = 0.010) and distance (*p* = 0.006) were the main factors in agricultural lands. There were no significant correlations in riverine wetlands. The correlation between the distance from the built-up urban areas and pollutant concentration showed that the closer the distance, the greater the concentration of PAHs.

## 1. Introduction

Polycyclic aromatic hydrocarbons (PAHs) are a group of ubiquitous persistent organic pollutants that are composed of two or more fused aromatic rings [1,2]. PAHs cause serious global environmental concerns for both ecosystems and human health because of their potential toxicity and carcinogenicity [3,4]. Urbanization is a current land use trend. Global urban areas have increased considerably in the last three decades [5]. Many phenomena come from urbanization, such as population aggregation, industrial development, increased construction of roads and vehicle use, and the construction of landfills [2]. Polycyclic Aromatic Hydrocarbons (PAHs) are a group of organic pollutants that are strongly related to anthropogenic activities such as settlement, transport, and industrial development [6,7]. Owing to rapid urbanization, the PAHs contamination to the environment is accelerating, which increases the potential harm to human health [8]. PAHs in urban areas can be emitted into urban rivers through multiple pathways, including wastewater discharge, atmospheric deposition, oil spillage, and surface run off [9]. Due to their lipophilicity and persistence, PAHs tend to persist in the soil [10]. After deposition on surface soil, PAHs may further accumulate in vegetables and other biota and be transfered to humans via the food chain [11,12], or they can strongly sorb on soil, where they persist for long periods of time [7]. Once they enter the urban river system, PAHs are easily adsorbed in particulate matter and finally deposited in the sediment. Sedimentary PAHs can release into the overlying water and continue to threaten urban river ecosystems and human health [7,13]. After a range of biochemical reactions, sediments/soils are the final deposition sinks of PAHs.

Several studies have described the occurrence of PAHs in sediments/soils. Studies have shown different correlations between the concentration of PAHs, particle size, total organic carbon (TOC), and other physiochemical factors in sediments/soils [13,14,15,16,17]. Recent studies have shown that other factors such as land use type, population density, distance from population centers, and urbanization history have a significant effect on the distribution of soil/sediment PAHs [13,18,19]. The distribution of PAHs in sediments/soils could possibly be affected by both sediment/soil characteristics and factors associated with urbanization, but neither relationship is well resolved. To evaluate the potential risk caused by PAHs to river and lake aquatic ecology, we calculated B*a*P-based total toxic equivalency (TEQ-B*a*P) [19] and toxic unit (TU) [20,21] metrics based upon individual concentration.

Chaohu Lake is located in the center of the Anhui Province, Eastern China, between the Yangtze and Huaihe rivers. It is the fifth largest freshwater lake in China. The Shiwuli River is located in the Binhu New District; it serves as the principal source of pollutants flowing into Chaohu Lake. Binhu New District is in the process of rapid urbanization. Half of the water source of the Shiwuli River currently comes from the tail water of the sewage treatment plant [22]. The total design scale of the treatment plant is 100,000 tons of sewage per day, and the planned total service area is 131.4 square kilometers. Due to the increasing input of pollutants linked with urbanization, large amounts of PAHs are currently transported to the lower reaches of the Shiwuli River. Pollutants are discharged from original major industrial pollution sources, such as the Jianghuai chemical fertilizer plant and the Hefei chemical fertilizer plant (Hongsifang Chemical Group), other industrial enterprise pollution originates from industrial parks, non-point source pollution caused by urban road traffic, and agricultural non-point source pollution [22]. Along with the rapid increase in population during the process of urbanization, and the increase in impervious surfaces across the landscape, the original mechanisms of the natural hydrological cycle have been changed, and the pollution loads in surface runoff feeding any receiving water body have been intensified [23]. The upper and middle reaches of the Shiwuli River are built-up urban areas. The lower reaches of the Shiwuli River are downstream of the intersection of the Shiwuli and Baohe Avenue (Figure 1). The lower reaches flow through the suburbs [24] and are an ideal area to investigate PAHs pollution associated with urbanization activities.

The primary objectives of this study were to: (1) Characterize the PAH pollution level and composition using 30 cm sediment/soil cores from the lower reaches of the Shiwuli River of Chaohu lake, (2) assess potential ecological risks posed by PAHs in the sediments/soils, and (3) analyze the key factors that influence the risk and distribution of sediment/soil PAHs. Our study will contribute to strengthening the understanding of PAH distribution characteristics in sediments/soils, and thereby contribute to the implementation of effective pollution mitigation strategies for areas affected by urbanization.

## 2. Materials and Methods 

### 2.1. Sample Collection

We extracted twelve columnar soil samples from sites representing three land use types (ditch wetland, agricultural land, and riverine wetland) (Table 1) with a TC-600H piston columnar mud collector (Qingdao Daneng Environmental Protection Equipment Co., Ltd., Qingdao, China). Cores were collected from the lower reaches of the Shiwuli River in Chaohu Lake, China, from July 20–27, 2016. In order to reveal the effect of urban activity on soil pollution, core sampling was conducted in triplicate for each sampling site. Sampling sites were selected and referenced relative to the location where the Shiwuli River crosses Baohe Avenue, which is the border between the urban and rural areas of Hefei City (Figure 1). Sites 1, 2, and 3 were 2 km downstream of the Shiwuli-Baohe cross (117°19′ N and 31°45′ E). Sites 4, 5, and 6 were 4 km downstream of the cross (117°21′ N and 31°45′ E), and sites 7, 8, and 9 were 6 km downstream (117°22′ N and 31°43′ E). Lastly, sites 10, 11, and 12 were 8 km downstream of the cross (117°23′ N and 31°44′ E). Every sediment/soil core from each site was stratified into six layers (0–5 cm, 5–10 cm, 10–15 cm, 15–20 cm, 20–25 cm, 25–30 cm). We collected a total of 36 cores (12 sites × 3 cores), and the same layers of the triplicate cores were mixed together, creating a composite core sample for each depth, for each sampling site. All samples were placed in polyethylene bags, transported on ice, delivered to the laboratory within 24 h, and stored at −20 °C until analysis.

### 2.2. PAHs Extraction and Analysis

All freeze-dried samples were ground and passed through a 150 micron mesh sieve. We then extracted the PAHs using an accelerated solvent extraction solution (ASE-350, Dionex Company, Sunnyvale, CA, USA). Briefly, 5 g samples were mixed with 1 g diatomite and PAH standards of 20 μL 1 mg/L, and were then poured into a 34 mL extraction vessel. The extraction solvent was a 20 mL mixture of *n*-hexane and dichloromethane with a 1:1 volume ratio. The extraction pressure was 1500 psi and the temperature was 100 °C. The heating and static extraction times were 8 min and 10 min, respectively. The flushing volume was set at 60%, and the nitrogen purging time was 60 s. Three cycles were carried out. The extract was concentrated to 15 mL on a RE-52A rotary evaporator (Yarong Technology Co., Ltd., Fujian, China), purified using sulfuric acid and desulfurized using copper powder, concentrated to 2 mL in the rotary evaporator, and then purified with a composite silica gel purification column. The purification column was filled with 2 g of deactivated silica gel, 1 g acidic silica gel, 1 g deactivated silica gel, 1 g alkaline silica gel, 2 g activated silica gel and 2 g anhydrous sodium sulfate from bottom to top. PAHs were eluted with a mixture of 100 mL *n*-hexane and dichloromethane (1:1, V/V). The eluent was collected and concentrated to 2 mL in the rotary evaporator. Using a nitrogen sweeper, the eluent was slowly blown to nearly dry, with a constant volume of 300 mL.

The PAHs were analyzed by gas chromatography mass spectrometry (GC-MS, Agilent7890A/5975C, Agilent Technologies Inc., CA, USA) with a DB-5MS capillary column (0.25 mm × 0.25 µm × 30 m J&K Scientific, San Jose, CA, USA). The carrier gas was high purity He (99.999%) with a 1.0 mL/min flow rate. A sample (1 μL) was injected using an automatic sampling system. The temperatures of the inlet, ion source, and transmission lines were 250 °C, 240 °C, and 280 °C, respectively. The oven temperature was initially set at 80 °C for 2 min, then gradually increased (15 °C/min) to 215 °C for 1 min. The oven temperature was further increased (6 °C/min) to 280 °C for 1 min, and finally reached 300 °C at 10 °C/min for 5 min. The GS-MS detector was operated in electron impact mode at 70 eV with an ionic source temperature of 240 °C. A selected ion mode was used to identify the PAHs. The concentrations of the PAHs are given as the dry mass of the sample.

### 2.3. Organic Matter and Texture Analysis

Soil total organic carbon (TOC) was determined by the potassium dichromate oxidation external heating method according to the national standard (GB 7857-1987 and LY/T 1237-1999). Soil texture was determined by the pipette method (LY/T 1225-1999).

### 2.4. References and Reagents

Major reagents included the mixed standard solution of 16 priority PAHs: Naphthalene (Nap), Acenaphthylene (Acy), Acenaphthene (Acp), Fluorene (FLR), Phenanthrene (PHE), Anthracene (Ant), Fluoranthene (FLT), Pyrene (PYR), Chrysene (CHR), Benzo [*a*] anthracene (BaA), benzo[*k*]fluoranthene (BkF), Benzo[*a*]pyrene (B*a*P), Benzo[*b*]fluoranthene (BbF), Dibenz[*a,h*]anthracene (DhA), Benzo[*ghi*]perylene (BgP), and Indeno[1,2,3,*cd*]pyrene (IcP). The mixed standard solution was purchased from AccuStandard Company, New Haven, CT, USA. Chromatography-grade *n*-hexane and dichloromethane as an extract and eluent of PAHs were purchased from TEDIA Company, Fairfield, OH, USA. The anhydrous Na_2_SO_4_ was purified and activated prior to analysis in a muffle furnace at 400 °C for 4 h. The internal standard compound, Philippine-*d*_10_, was purchased from the J & K Company (San Jose, CA, USA). The internal standard compound was a pure substance, and the contents of the measured components were determined by comparison. Copper powder and silica gel were purchased from the National Pharmaceutical Group (Beijing, China). All other chemicals were analytical grade.

### 2.5. Quality Control 

After each triplicate sample group (*n* = 3) analysis, a blank sample was analyzed to monitor interference during sample processing. The analyte concentration of blank samples was less than 5% of the minimum concentration of all samples, representing the background signal due to sample extraction and measurement procedures. The limit of quantitation (LOQ) was defined as ten times that of the signal to noise ratio (10 S/N). In this study, the quantitative limit of PAHs in the surface sediment samples was obtained by adding the standard to 5 g sediment (dry weight). The LOQ of the PAH compounds ranged from 0.015 ng/g to 0.095 ng/g. The recoveries of the 16 PAH species ranged from 73.5% to 116.8% and the recovery of the internal standard compounds ranged from 86.3% to 108.4%. The calibration curves were plotted by peak area versus the concentrations of PAH compounds with R^2^ values ranging from 0.990 to 0.998 for the 16 examined PAHs.

### 2.6. Environmental Risk Assessment Approach

The B*a*P-based total toxic equivalency (TEQ-B*a*P) and PAH toxic units (TUs) were metrics employed to evaluate the ecological risk of each PAH compound to aquatic environments [25]. The toxic equivalency factor (TEF) method was developed to evaluate structurally-related compounds, sharing a common mechanism of action [20]. PAH toxic unit levels come from the United States Environmental Protection Agency (2003) procedures for the derivation of equilibrium partitioning sediment benchmarks (ESBs) for PAH mixtures [21]. 

TEQ-B*a*P values were calculated with the following equations based on the individual species concentrations of PAHs:TEQ − B*a*P*_i_* = ∑TEF*_i_* × C*_i_*(1)
ER = ∑TEQ − B*a*P_i_(2)
where TEF*_i_* represents the toxic equivalent factor of the PAH (Appendix A), C*_i_* is the concentration of contaminant *i* (in ng/g), and TEQ-B*a*P is calculated using the 16 priority PAHs proposed by the U.S. EPA that was selected in this study, with individual values of the PAHs provided in Appendix A. Total sample toxic equivalency, or ΣTEQ-B*a*P, was the sum of TEQ-B*a*P values measured for all contaminants in a sample unit. TU values were calculated with the following equations based on the individual species concentrations of PAHs:(3)$${\mathrm{TU}}_{i}=\frac{C_{i}}{\mathrm{TOC}\times \mathrm{Sediment}\;\mathrm{benchmark}}$$
TU = ∑TU*i*(4)
where C*_i_* is the concentration of contaminant *i* (in ng/g), TOC is the measured value of total organic soil carbon, equivalent sediment benchmark, or ESB*_i_* of contaminant *i* (Appendix A). Total sample toxicity ΣTU is the sum of the TU values measured for all contaminants in a sample unit.

### 2.7. Statistical Analysis

Non-metric multidimensional scaling (NMDS) and analysis of similarity (ANOSIM; permutations = 999) were executed with the vegan package (Version 2.0-2, Free Software Foundation, Inc., Boston, USA) of R v.2.8.1 project. The differences of the levels of PAHs among the three land use types were investigated using ANOSIM with R software (2.15.2, R Core Team, Vienna, Austria). We used Sigmaplot software to show the results of the analysis. To identify the relationship among PAHs and selected environmental factors, we performed a multivariate redundancy analysis (RDA) based on a Monte Carlo permutation to explore the correlations of the environmental properties with the spatial variability of PAHs. The software CANOCO (version 4.5, Microcomputer Power, NY, USA) was used to perform the RDA.

## 3. Results

### 3.1. Occurrence of PAHs in Surface Sediments/Soils

All 16 kinds of PAHs were detected in every surface sediment/soil. The total concentrations of PAHs, which refer to the group average, ranged from 36.5–1031.8 ng/g, with the highest value in the Baohe ditch wetlands. Overall, the total PAHs in the surface sediments/soils followed this order: ditch wetlands (617.2 ng/g group average), riverine wetlands (282.1 ng/g group average), agricultural lands (103.7 ng/g group average). The results showed that the concentrations of PAHs measured in this study were lower than other wetlands in China, such as the Taihu Lake estuary, the Liaohe estuary, and the Mawan mangrove wetland, but higher than that of the Baiyangdian wetland [26]. The pollution level of PAHs in the surface layer of the Shiwuli River lower reaches of Chaohu Lake was in the middle level compared with similar wetlands in China and farther afield (Table 2).

In this study, the maximum PAH concentration (1031.8 ng/g) was 30 times higher than the minimum value (36.5 ng/g). The content of PAHs in the surface layer (0–5 cm layer) of the ditch wetlands was 686.3 ± 195.9 ng/g. The surface PAHs content of the riverine wetlands was 283.5 ± 203 ng/g, and the content of PAHs in the surface soils of the agricultural lands was 105.1 ± 47.8 ng/g. 

### 3.2. Vertical Distribution Characteristics of PAHs 

The spatial distribution of PAHs was analyzed in different land use types. The patterns of vertical PAH distribution were significantly different among different land use types (Table 3). The same types of land use gather together, and different types of land use show larger distances (Figure 2).

To further understand the spatial distribution of PAHs, we investigated the composition and analyzed the differences of PAH species in different soil depths. PHE, Ant, FLT, PYR, CHR, and BkF were the main pollutants in ditch wetlands, accounting for about 65% of the total concentration, independent of depth. Napm PHE, Ant, and FLT were the main pollutants in agricultural lands, accounting for approximately 65% of the total concentration, independent of depth. PHE, FLT, PYR, CHR, BgP, and IcP were the main pollutants in riverine wetlands, accounting for about 57% of the total concentration, independent of depth. PHE and FLT were the main pollutants in the lower reaches of the Shiwuli River. PAHs in the ditch wetlands were mainly distributed evenly throughout the soil profile, and PAHs in agricultural land soils were mainly aggregated in the 0–20 cm layer. The low-cyclic PAHs (LPAHs) in the riverine wetlands had a uniform distribution and varied only slightly throughout the sediment cores, while the high-cyclic PAHs (HPAHs) were mainly concentrated in the 0–20 cm layer (Table 4). Low-cyclic PAHs (LPAHs) include 2- and 3-ring PAHs and high-cyclic PAHs (HPAHs) include 4-, 5- and 6-ring PAHs. It has been reported that LPAHs (less than 4 rings) are mainly generated by low- or moderate-temperature combustion processes (i.e., biomass combustion and domestic coal burning), while HPAHs (ring numbers greater than 3) are mainly generated by high-temperature combustion processes (i.e., vehicular exhaust and industrial coal combustion) [33].

### 3.3. PAHs and Environmental Factors

The upper and middle reaches of the Shiwuli River are built-up urban areas. Below Baohe Avenue (Figure 1) are the lower reaches of the Shiwuli River, and the lower reaches flow through the suburbs [24]. We defined sampling point distance as measured from the cross of the Shiwuli River and Baohe Avenue. There was a relationship between the distance and PAH composition in all three land use types. Sampling locations were grouped into four groups (Figure 3); the distance of the four groups from the cross of Baohe Avenue was 2 km, 4 km, 6 km, and 8 km.

### 3.4. Environmental Risk Assessment

Ecological risk (ER) was evaluated by two models based upon benzo[*a*]pyrene toxic equivalency (TEQ-B*a*P) and total toxic units (TUs). ER assessments were made utilizing the sediment quality guidelines for total PAHs [25]. Effects range low (ERL) and the effects range median (ERM) values of 16 PAHs in the aquatic sediment [34] were further computed by TEF (toxic equivalent factor) values to obtain TEF-adjusted TEQ-B*a*P (ERL) and TEQ-B*a*P (ERM) values [25]. In our study, the TUs of PAHs ranged from 0.11–0.28 for all samples at different depths of the three land use types (Figure 4), which might not cause mortality of local benthic organisms.

### 3.5. Land Use Effects on PAH Composition 

#### 3.5.1. Ditch Wetlands

The correlation between PAH composition and environmental factors among different sampling points is illustrated in a hybrid RDA correlation biplot map (Figure 5A). Partial RDAs based on a Monte Carlo permutation (*n* = 499) kept only the significant parameters in the models, indicating that these environmental factors might be important for explaining the PAHs compositions. TOC (*p* = 0.01), clay (*p* = 0.02), and distance (*p* = 0.02) were more important in affecting the PAH compositions compared to other environmental factors (Table 5). These three statistically significant (*p* < 0.05) variables explained 43% of the total variation in PAHs composition (Figure 5A) in the ditch wetlands. In contrast, silt (*p* = 0.07), depth (*p* = 0.31), and sand (*p* = 0.37) have no significant correlations with the PAHs. The first and second axes explained 36.2% and 16.9% of the total variance, respectively, for the ditch wetlands.

#### 3.5.2. Agricultural Lands

Two significant (*p* < 0.05) variables explained 37% of the total variation in PAH composition (Figure 5B) in agricultural lands. Depth (*p* = 0.01) and distance (*p* = 0.006) were more influential on the PAHs composition compared to other environmental factors (Table 5). Clay (*p* = 0.366), TOC (*p* = 0.796), sand (*p* = 0.974), and silt (*p* = 0.978) did not have significant correlations with the PAHs concentrations. In agricultural lands, the first axis explained 26.7% of the total variance and the second axis explained 4.9% of the total variance for the agricultural lands.

#### 3.5.3. Riverine Wetlands

In riverine wetlands, there were five variables that explained 34% of the total variation in PAH composition (Figure 5C). Sand (*p* = 0.128), TOC (*p* = 0.25), distance (*p* = 0.122), depth (*p* = 0.136), and silt (*p* = 0.27) all affected the composition of pollutants, but the effects were not significant (Table 5). The first axis explained 23.8% of the total variance. The second axis explained 5.4% of the total variance for the riverine wetlands. 

## 4. Discussion

According to the classification established by Maliszewska-Kordybach (Table 6) [35], all surface sediments in the ditch wetlands in this study were contaminated. In particular, the ditch wetlands of Baohe were heavily contaminated, which may be due to the sewage discharge. The samples were collected where the river course was straight and the riverbed was high, resulting in poor water mobility and the accumulation of PAHs in surface sediments. Meanwhile, other studies have shown that approximately half of the water in the Shiwuli River during the dry season and flat water period, when rivers are at normal water levels, come from the tail water of the sewage treatment plant [22]. Although treated, this tail water conveys the industrial, domestic, and agricultural drainage that contains pollutants. Furthermore, the sampling site of the Baohe ditch wetland was the sampling site closest to the tail water discharge point. The ditch wetland site was most affected by contamination from the sewage treatment plant.

All benzo[*a*]pyrene toxic equivalent (TEQ-B*a*P) values of the PAHs in the sediments/soils were lower than the low range of the collective TEQ-B*a*P (effects range low, or ERL). The ERL in this study was obtained from the literature [25]. The distributions of the toxicity effect data were determined using percentiles. The lower 10th percentile of the effects data for each chemical was identified and referred to as the effects range low (ERL) [34]. Furthermore, the maximum TEQ-B*a*P value of this study was 369, which was less than the ERL (533). Mart (2007) reported a mean amphipod mortality of >9% in samples where the sum of the TUs of PAHs exceeded 0.1. Moreover, they reported a mean amphipod mortality of >30% where the sum of the TUs of PAHs exceeded 0.5, and >67% mortality where the TUs of PAHs exceeded 1.0. TUs can be used as a useful indicator of ecotoxicity [25]. The environmental risk was higher in ditch wetlands than in agricultural lands and riverine wetlands. Our TUs and TEQ-B*a*P values indicated that all three types of wetlands were at a low ecological risk. We suggest three reasons for this low ecological risk. Firstly, the relocation of polluting enterprises in the upper reaches of Shiwuli River, which could reduce the pollution emissions from upstream. Secondly, the treatment effect of the Shiwuli River sewage treatment plant, as surface runoff sewage and domestic sewage are discharged into sewage treatment plants, causes the amount of pollutants entering the rivers to be reduced. Lastly, the lower reach of the Shiwuli River is surrounded by farmland and villages with less pollutant emissions. These three factors could lead to a reduction of wastewater discharge and PAHs downstream of the Shiwuli River.

The PHE and PYR PAH species both had good fit effects on the TUs of ditch wetlands. There was a significant positive correlation between TUs and PHE (*R^2^* = 0.530, *p* < 0.01). TUs were negatively correlated with PYR (*R^2^* = 0.530, *p* < 0.01) (Table 7). In riverine wetlands, there was no significant correlation between the concentration of PAHs and TUs. Obviously, PAH accumulation in the riparian soils is not only controlled by upstream processes but also by the adjacent land use types. Furthermore, PAHs can exchange between river water and floodplain sediments. The complexity of PAHs sources might lead to the complexity of TUs. In agricultural lands, FLT and DhA had a good fitting effect on TUs. There was a significant positive correlation between TUs and FLT (*R^2^* = 0.522, *p* < 0.01). TUs were negatively correlated with DhA (*R^2^* = 0.522, *p* < 0.01) (Table 7). 

The toxicity of high ring compounds such as BbF, BkF, Inp and BgP is strong. Recent studies showed that the relative toxicities of the five PAHs were BkF, IcP, B*a*P, BbF, CHR. The contribution of these five PAHs to the overall toxicity of PAHs in sediment/soil samples of Taihu Lake was more than 50% [36]. BbF, BkF, Inp and BgP are carcinogenic. The interaction of these PAHs further strengthens their carcinogenicity [37,38,39]. These compounds were detected in our sediment/soil samples. Due to a lack of minimum safety values [34], the presence of these compounds in sediments/soils still has certain ecological risks.

There were no obvious changes in the concentrations of the 16 PAHs by depth in the ditch wetlands. For sediment cores from the ditch wetlands, the relative percentage of individual PAH species showed minor variations in composition at different depths. The diagnostic ratio is a widely used technique to apportion the origin and sources of PAHs present in different environmental media. The PHE/Ant ratio (<10) and HMW/LMW (>1) at different depths indicate a pyrogenic origin [40]. The diagnostic ratio shows similar origins and sedimentary processes for PAH compounds [25]. It is also possible that the widening of rivers in recent years has led to disturbances in the deep sediments, which may have affected the normal distribution of PAHs with increasing depth.

We observed significant positive correlations between the 16 PAHs and TOC (*p* = 0.01); this is especially true of the relationship between HPAHs (i.e., >4 ring PAHs) and TOC in ditch wetlands. This results in a higher correlation coefficient between TOC and HPAHs than TOC and LPAHs (i.e., 2-ring + 3-ring PAHs). This result could be related to the high concentration of TOC and the strong adsorption with HPAHs [41]. TOC and PAHs can also precipitate or co-occur in wastewater [15], as previously demonstrated when treated municipal sewages were found to be the major sources of the water of the Shiwuli River [22]. Due to the limitations of sewage treatment technology, TOC and PAHs cannot be removed effectively, which is likely the reason that PAHs and TOC were positively correlated in ditch wetland soils.

A significant correlation between the clay load and the concentration of the 16 PAHs was found (*p* = 0.02) in the ditch wetlands. The correlation with silt and sand was not significant. In this study, LPAH and HPAH were enriched in fine particles. These findings could have resulted from three factors. First, some major sources of PAHs produced particulates with different particle sizes. In other words, PAHs were produced by particles of a specific diameter. Second, finer particles with high specific surface areas could enhance the adsorption of PAHs [15]. Third, the lower reaches of the Shiwuli River were straight, and the riverbed may have been silted up for a long time, resulting in poor water flow. Therefore, the soil was mainly composed of silt and clay fractions, which have larger specific surface areas and a higher adsorption capacity of elements [42]. This might also be beneficial for the enrichment of PAHs in clay within the ditch wetlands.

The concentrations of 16 PAHs in riverine wetlands decreased with increasing depth. In particular, the concentrations of the 5- and 6-ring PAHs decreased by a minimum of two and a maximum of ten times with depth, which indicated that HPAHs might be absorbed by particulate matter, resulting in a smaller radial vertical distribution. Thus, if enriched on the surface, there is a lower concentration on the bottom. These results indicated that the main HPAHs were distributed in the surface layer. Compared with HPAHs, the strong vertical migration of LPAHs promoted the radial distribution of LPAHs [43,44]. The result indicated a uniform distribution of LPAHs, which varied only slightly between the sediment cores in the riverine wetlands.

Riverine wetland soils are mainly formed by the periodic deposition of suspended sediments from river water during flood events [45]. However, the PAH accumulation in the riverine wetlands was not only controlled by upstream processes but also by the adjacent or upland land uses. For example, riverine wetlands adjacent to agricultural fields could be influenced by mechanical operations, misplaced fertilizer, and/or pesticide drift. Riverine wetlands adjacent to industrial and commercial regions could be influenced by household garbage and wastewater. These pollutants may be ultimately deposited in wetlands, which could lead to changes in the chemistry of certain riverine wetlands [46]. Complicated and variable sources of PAHs might cause weak or insignificant correlations between PAHs and environmental factors.

The concentration of LPAHs was 6–17 times higher than that of HPAHs within the same horizons of agricultural land soils, which was different from the ratio (1–2 times) in ditch wetlands and riverine wetlands. This might indicate that the sources of PAHs in agricultural lands were different from those in ditch and riverine wetlands. For example, agricultural soils had better adsorption and an enrichment of LPAHs (2- and 3-ring PAHs) [47]. Due to the higher biodegradability, solubility, and volatility, and lower sorption ability compared to HPAHs, LPAH levels accumulated in the soil were lower than HPAH levels. However, the higher presence of 2- and 3-ring PAHs could suggest that more PAHs came from recent deposition [48]. In addition, the result shows that PAHs in agricultural lands were mainly found in the surface aggregates (Table 4). Studies have shown that plant roots can absorb PAHs and transport LPAHs to the surface, and that HPAHs can attach to roots [49]. Leaves and other plant tissues can absorb atmospheric PAHs [50], which can be deposited into the surface soil via rainwater and deciduous leaves [51]. Therefore, plant–PAH interactions could contribute to the shallower distribution of PAH. Because of long-term or intermittent flooding, there is less vegetation in ditch wetlands and riverine wetlands, so the influence of vegetation on the distribution of PAHs is not obvious compared with agricultural lands [49,50]. 

The upper reaches of the Shiwuli River from Baohe Avenue were in urban built-up areas. The reaches below the Baohe Avenue were in agricultural areas. This distance was approximately equal to the distance from the sampling point to the urban built-up area. With increasing distance from the area of rapid urbanization, the concentration of PAHs in the sediment/soil gradually decreased. One potential source of PAHs is urban areas and vehicular transportation. Through further analysis of the correlations between the distance of urban areas and the residual level of PAHs in different land use types, we found that the closer the distance, the greater the concentration of pollutants, independent of land type and sediment/soil properties. Our study results are similar to the results of Zheng et al. [18]. PAHs in urban areas come mainly from anthropogenic activities [9]. Urbanization and industrialization have led to deterioration of environmental quality. This may be explained by the fact that urban areas have an impact on the lower reaches of the Shiwuli River which could overshadow the effects of TOC and grain size. Distance from the urban areas is a very important factor affecting the distribution of PAHs in the lower reach of Shiwuli River. 

## 5. Conclusions

The concentrations of the priority control 16 PAHs in the surface sediments/soil varied from 36.5 ng/g to 1031.8 ng/g. The total PAHs in the surface sediments/soils followed this order: ditch wetlands (617.2 ng/g group average), riverine wetlands (282.1 ng/g group average), agricultural lands (103.7 ng/g group average). All surface sediments in the ditch wetlands in this study were contaminated by PAHs. Vertical distributions of PAHs in the top 30 cm of sediments/soils of different land use types were significantly different. In ditch wetlands, PAHs were distributed fairly evenly throughout the sediment profile (top 30 cm), while PAHs in agricultural soils were mainly surface aggregates. The LPAH concentrations in riverine wetlands had a uniform distribution and varied only slightly between soil cores, whereas the HPAHs were concentrated in the surface layer. This study showed that the relationships between PAH concentrations and sediment/soil properties were inconsistent in three land use types. Through further analysis of the correlation between the distance between sampling points and the built-up area and residual levels of PAHs, we found that the closer the distance from the built-up area, the greater the concentration of PAHs. This pattern was independent of land use type and sediment/soil properties. 

The environmental risk was higher in the ditch wetlands than in the agricultural lands and riverine wetlands. We also detected high-toxicity elements, such as BbF, BkF, InP and BgP, which do not have minimum safety values. The presence of these compounds in sediments/soils conveys certain ecological risks.

## Figures and Tables

**Figure 1 ijerph-16-02302-f001:**
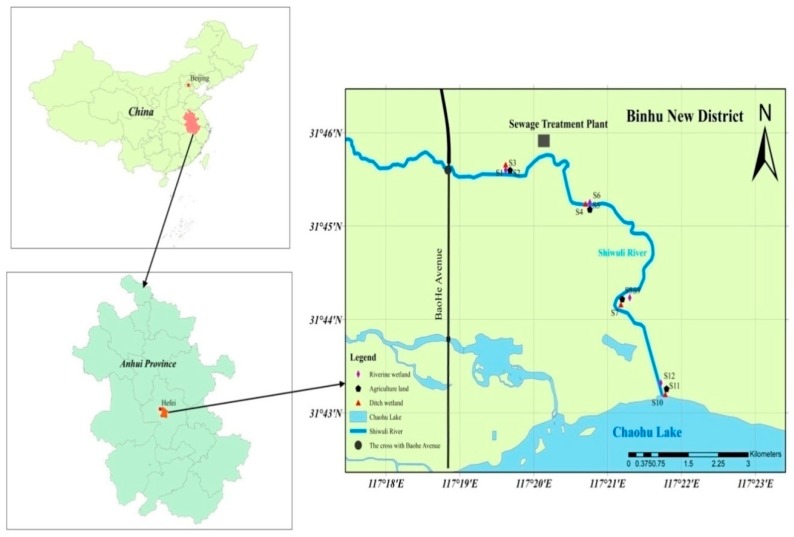
Sampling sites and regional context.

**Figure 2 ijerph-16-02302-f002:**
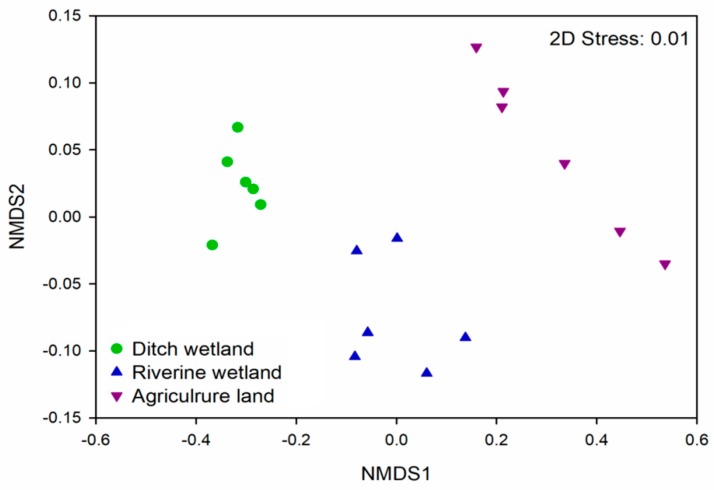
Non-metric multidimensional scaling plot showing the differences of PAHs in three land use types.

**Figure 3 ijerph-16-02302-f003:**
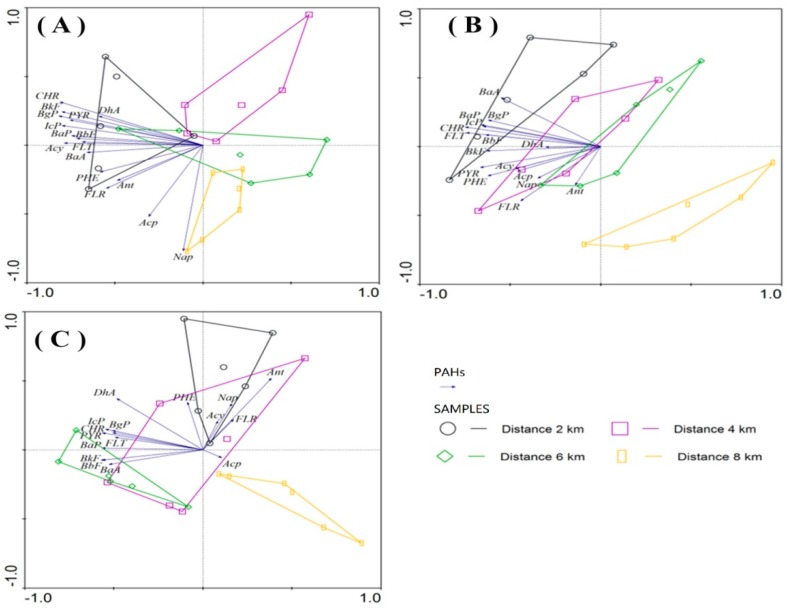
Redundancy analysis (RDA) of the correlation between distance (from urbanization) and the PAHs concentrations in ditch wetlands (**A**), agricultural lands (**B**), and riverine wetlands (**C**). The black group represents a distance of 2 km, the purple group represents a distance of 4 km, the green group represents a distance of 6 km, and the orange group represents a distance of 8 km.

**Figure 4 ijerph-16-02302-f004:**
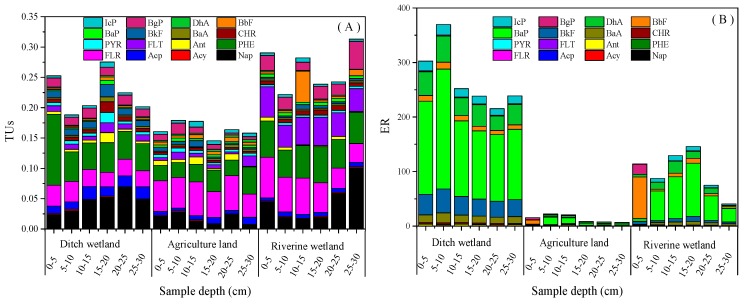
Total toxic units (**A**) and environmental risk of benzo[*a*]pyrene toxic equivalent (TEQ-B*a*P) (**B**) of PAHs in sediment/soil cores by depth (0–5 cm sediment/soil layer, 5–10 cm sediment/soil layer, 10–15 cm sediment/soil layer, 15–20 cm sediment/soil layer, 20–25 cm sediment/soil layer, and 25–30 cm sediment/soil layer in ditch wetlands, agricultural lands, and riverine wetlands).

**Figure 5 ijerph-16-02302-f005:**
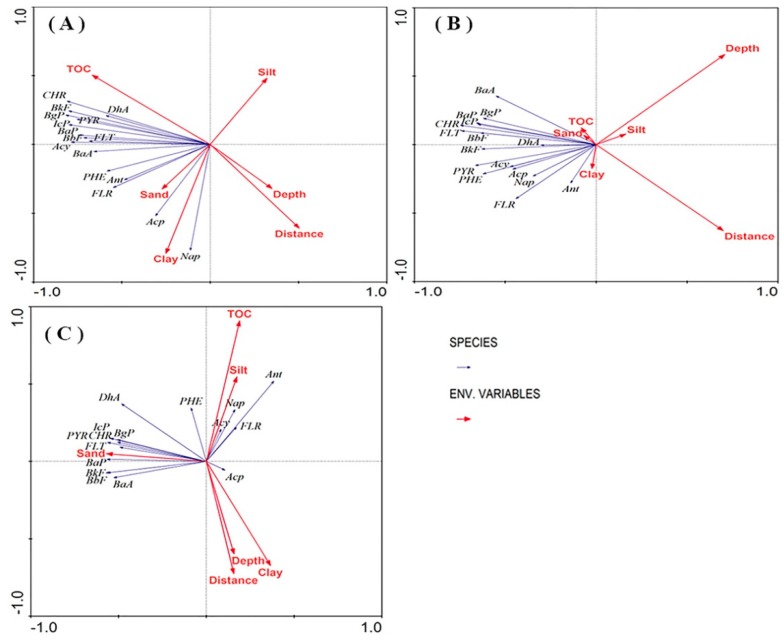
Biplot map of hybrid redundancy analysis (RDA) for PAHs and soil properties. The blue arrows represent PAHs and the red arrows represent the influencing factors. The length of the arrow represents the degree of influence for each environmental factor on the PAHs composition in ditch wetlands (**A**), agricultural lands (**B**), and riverine wetlands (**C**).

**Table 1 ijerph-16-02302-t001:** Main properties of the sediments/soils of the Shiwuli River.

Sampling Site		Distance from the Cross with Baohe Avenue (km)	Main Properties	Land Use Type
Name	Number			
Baohe	S1	2	Covered by water	Ditch wetland
	S2	2	Dominated by crops	Agricultural land
	S3	2	Dominated by wetland vegetation or indigenous shrubs and grasslands	Riverine wetland
Yicheng	S4	4	Covered by water	Ditch wetland
	S5	4	Dominated by crops	Agricultural land
	S6	4	Dominated by wetland vegetation or indigenous shrubs and grasslands	Riverine wetland
Xiwang Bridge	S7	6	Covered by water	Ditch wetland
	S8	6	Dominated by crops	Agricultural land
	S9	6	Dominated by wetland vegetation or indigenous shrubs and grasslands	Riverine wetland
Estuary	S10	8	Covered by water	Ditch wetland
	S11	8	Dominated by crops	Agricultural land
	S12	8	Dominated by wetland vegetation or indigenous shrubs and grasslands	Riverine wetland

**Table 2 ijerph-16-02302-t002:** Concentration of surface sediments/soils polycyclic aromatic hydrocarbons (PAHs) in different regions of the world (ng/g).

Sampling Locations	Levels/Range (ng/g)	Reference
Taihu Lake estuary sediment, China	371–2530	[27]
Liaohe estuary wetland, China	293.4–1735.9	[28]
Baiyangdian wetland, China	146.0–645.9	[26]
Mawan mangrove wetland, China	1300–5000	[29]
Anzali wetland, Iran	212.0–2674.0	[30]
Dishui Lake wetland, China	11.49–157.09	[31]
Todos Santos Bay wetlands, Mexico	96	[32]
Sediments/soils in the lower reaches of Shiwuli River, China	36.5–1031.8	This study

**Table 3 ijerph-16-02302-t003:** The differences of PAHs in three land use types based on the similarity test of analysis of similarity (ANOSIM).

Treatment	ANOSIM	
	*r*	*p*
Ditch wetland vs. Riverine wetland	0.985	0.005
Ditch wetland vs. Agricultural land	1	0.005
Riverine wetland vs. Agricultural land	0.8	0.003

**Table 4 ijerph-16-02302-t004:** Composition of PAHs in different depths of three land use types.

Land Use Type	PAHs (ng/g)	Depth (cm)					
		0–5	5–10	10–15	15–20	20–25	25–30
Ditch wetland	2,3-ring PAHs	203.79	151.50	201.59	175.90	155.10	163.41
	4-ring PAHs	160.66	256.67	225.99	198.70	175.03	182.89
	5,6-ring PAHs	261.33	300.23	226.29	209.39	190.44	209.37
Riverine wetland	2,3-ring PAHs	100.71	85.99	60.78	55.46	49.67	50.57
	4-ring PAHs	87.77	63.59	84.83	96.36	56.94	45.34
	5,6-ring PAHs	95.05	68.01	104.03	114.31	59.76	36.99
Agriculture land	2,3-ring PAHs	71.43	69.13	89.86	30.13	52.62	20.61
	4-ring PAHs	21.22	21.87	20.61	11.80	10.23	10.42
	5,6-ring PAHs	12.41	15.51	15.34	5.84	4.88	4.69

**Table 5 ijerph-16-02302-t005:** RDA results showing the percentage variance explained for PAHs in three land use types.

Land Use Types	Explanatory Variables	%Variance Explained	*p* Value	*F* Ratio
Ditch wetland	TOC	21	0.01	5.78
	Clay	13	0.02	4.26
	Silt	8	0.07	2.64
	Distance	9	0.02	3.67
	Depth	3	0.31	1.07
	Sand	3	0.37	1.05
	**All factors**	**57**		
Agricultural wetlands	Depth	16	0.01	4.07
	Distance	15	0.006	4.59
	Clay	3	0.366	1.03
	TOC	2	0.796	0.42
	Sand	0	0.974	0.18
	Silt	1	0.978	0.11
	**All factors**	**37**		
Riverine wetland	Sand	9	0.128	2.11
	TOC	5	0.25	1.36
	Distance	8	0.122	2.01
	Depth	8	0.136	2.17
	Silt	4	0.27	1.17
	**All factors**	**34**		

Partial RDAs based on a Monte Carlo permutation (*n* = 499), keeping only the significant parameters in the models. *F* and *p* values were estimated using Monte Carlo permutations. The higher the *F* value, the stronger the effect of the variables.

**Table 6 ijerph-16-02302-t006:** Classification of soil contamination by PAHs [35].

Class of Soil Contamination	∑PAH (ng/g)
Not contaminated	<200
Weakly contaminated	200–600
Heavily contaminated	>1000

**Table 7 ijerph-16-02302-t007:** Multiple linear stepwise regression analysis between toxic units (TUs) and the concentration of PAHs in two land types.

Land Use Types	Regression Formula	Coefficient of Determination	*F* Value	*p* Value
Ditch wetland	TU = 33.33 × 10^−3^ + 0.490 × 10^−3^ PHE − 0.293 × 10^−3^ PYR	0.530	11.836	<0.01
Agricultural land	TU = 41.161 × 10^−3^ + 1.740 × 10^−3^ FLT − 13.889 × 10^−3^ DhA	0.522	11.458	<0.01

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
