# Peer review of "Polycyclic Aromatic Hydrocarbons in Sediments/Soils of the Rapidly Urbanized Lower Reaches of the River Chaohu, China"

_ijerph, 2019, doi:10.3390/ijerph16132302_

Round 1
Reviewer 1 Report
This is research on monitoring and assessment of the river sediment quality in river Chaohu. It a basically a case study of a very local interest. Authors should draw perhaps more universal conclusions in order to be scientifically important. They should compare their results to other similar ecosystems outside the China in order for this research to be more significant.
Other commends are as follows:
-Since the main objective of the research is to assess the level of contamination of river’s sediments with PAHs, thus it necessary for authors to address issue of possible sources of PAHs contamination in more detail. Now in the introduction section they just put literally one sentence on that subject, that is "due to the increasing input of pollutants with urbanization, large amount of PAHs are currently transported to the lower reaches of the Shiwuli River", which should be considered as insufficient. It is highly nonspecific to just point out “urbanization” as the source. Please be more detailed in determining point sources of pollution with PAHs.
- in materials and method in. 2.6 section, there are lacking references regarding used and applied methodology of assessing environmental risk using total toxic equivalency. Please explain also in one-two sentences the purposiveness and applicability of this specific methodology for this study.
- table 1 description is not presented in full (line 193), please name this table correctly.
- section- 3.1. is very poorly discussed. Authors should at least refer their results (total PAHs concentration) to some threshold values, or norms or regulations, background values etc. (if there is no such regulations, reference this values to other river sediments outside China, for the comparison). Moreover it will be valuable if authors at least mention which individual PAHs was contaminating the most this river’s sediment, especially since they discuss contamination with individual PAHs in discussion section (4).
Author Response
Dear Reviewer,
Thank you very much for reviewing our manuscript (ijerph-522967) and providing us with many constructive comments, all of which have improved the quality of our work. We have carefully considered the comments from the Reviewer, and we have revised the manuscript accordingly. The English language in the manuscript was polished and revised by LetPub, an author services company. In addition, we have carefully checked the entire manuscript to ensure that the formatting requirements meet the journal’s guidelines. Please find the revised manuscript as well as our responses (highlighted in red) to each of the Reviewer’s comments below.
Thank you for your time and consideration.
I wish you good health and much happiness.
Yours sincerely,
Huanling Wu
June 20, 2019
Point 1: This is research on monitoring and assessment of the river sediment quality in river Chaohu. It a basically a case study of a very local interest. Authors should draw perhaps more universal conclusions in order to be scientifically important. They should compare their results to other similar ecosystems outside the China in order for this research to be more significant.
Response 1: We appreciate this Reviewer’s positive comments on our work. We have carefully considered their comments. We compared the PAHs in the surface sediments/soils to other similar ecosystems in China and farther afield. Please see the revisions on page 6, lines 202–206. However, we note that there is no published comparisons of pollutant concentrations for all depths of columnar sediments/soils, as research on columnar sediments/soils is often divided into different profile lengths in different studies.
Point 2: -Since the main objective of the research is to assess the level of contamination of river’s sediments with PAHs, thus it necessary for authors to address issue of possible sources of PAHs contamination in more detail. Now in the introduction section they just put literally one sentence on that subject, that is "due to the increasing input of pollutants with urbanization, large amount of PAHs are currently transported to the lower reaches of the Shiwuli River", which should be considered as insufficient. It is highly nonspecific to just point out “urbanization” as the source. Please be more detailed in determining point sources of pollution with PAHs.
Response 2: We apologize for not being more specific in our initial draft. We agree with this Reviewer and have elaborated on this topic in detail. Please see the revisions on page 2, lines 72–78.
Point 3:in materials and method in. 2.6 section, there are lacking references regarding used and applied methodology of assessing environmental risk using total toxic equivalency. Please explain also in one-two sentences the purposiveness and applicability of this specific methodology for this study.
Response 3: Thank you for the helpful suggestion. Following the Reviewer’s comments, we have added the appropriate references and explained the purposiveness and applicability of this methodology. Please see page 5 for our amendments (lines 169–173).
Point 4: table 1 description is not presented in full (line 193), please name this table correctly.
Response 4: Thank you for this comment. The table title has been revised. Please see page 6, line207.
Point 5: section- 3.1. is very poorly discussed. Authors should at least refer their results (total PAHs concentration) to some threshold values, or norms or regulations, background values etc. (if there is no such regulations, reference this values to other river sediments outside China, for the comparison). Moreover it will be valuable if authors at least mention which individual PAHs was contaminating the most this river’s sediment, especially since they discuss contamination with individual PAHs in discussion section (4).
Response 5: We agree with this Reviewer. According to the range of PAHs concentrations provided by the previously published literature, the degree of pollution of PAHs in surface sediments was compared and classified. See page 12, lines 300–311. Additionally, I think it is valuable to identify the main pollutants. As a result, the main pollutants identified for the three land use types were added; please see the changes on page 7, lines 223–228.
Reviewer 2 Report
I think this work has merit to be published. I have only a few comments
L107, 116 and 140 Instead of "n-hexane", it should be used "n-hexane"
L152, 153 and 155 It is suggested to specify PAHs associated with those LOQ, recoveries and R values.
L182 I think that those are antecedents
L 422 Conclusions must be improved
Author Response
Dear Reviewer,
Thank you very much for reviewing our manuscript (ijerph-522967). We appreciate the reviewer’s positive comments on our work. We have carefully considered the comments from the Reviewer. We have revised the manuscript accordingly. The English language in the manuscript was polished and revised by LetPub, an author services company. In addition, we have carefully checked the entire manuscript to ensure that the formatting requirements meet the journal’s guidelines. Please find the revised manuscript as well as our responses (highlighted in red) to each of the Reviewer’s comments below.
Thank you for your time and consideration.
I wish you good health and much happiness.
Yours sincerely,
Huanling Wu
June 20, 2019
Point 1: I think this work has merit to be published. I have only a few comments
Response 1: We appreciate this Reviewer’s positive comments on our work.
Point 2: L107, 116 and 140 Instead of "n-hexane", it should be used "n-hexane"
Response 2: This formatting error has been corrected in the revised manuscript. Please see page 4, line 116 and line 125, and page 5, line 149.
Point 3: L152, 153 and 155 It is suggested to specify PAHs associated with those LOQ, recoveries and R values.
Response 3: Thanks for this helpful suggestion. We note that there are 16 kinds of PAHs, and LOQ, recoveries and R values are within a certain range. Since each test is not completely consistent, the values are not marked.
Point 4: L182 I think that those are antecedents
Response 4: After reviewing our manuscript, we decided that this result is not appropriate for this study. As such, this text has been deleted.
Point 5: L 422 Conclusions must be improved.
Response 5: We agree with the Reviewer and have revised the conclusions accordingly. We added a description of the range of PAHs concentrations in surface sediments and soils and also revised the order of presenting our important findings within this section. We hope that our revisions align with your suggestions.
Reviewer 3 Report
There are few studies that analyzes the relation with urbanization, although there are many papers on PAH distribution of sediment / soil, This work obtained several interesting results: the vertical PAH distribution was different depending on the use types; the environmental risk of PAHs was the highest in the ditch land; the PAH distribution depends on the build-up of urban areas, etc., which merit publication.
However, there are several points which need more consideration, clarification and typographical errors. They are described below.
Major comments:
1) Fig. 4 shows that the PAH distribution differed depending on the depth. Although the depth of sediment is an important parameter indicating the time, the relationship with the history of human activity has not been considered at all in Discussion.2) Also, the residue of PAH in sediments is considered to be different depending on the difference in the loading paths (inflow/falling), presence or absence of artificial work, decomposition reactions (microbe/photolysis) etc. However, Fig. 4 only describes that the type and concentration of PAH are different depending on the depth, and the relationship with differences in load paths, artificial work, decomposition reactions etc. is not considered in Discussion.
Minor comments:
Abstract, Fig. 4 caption and 4. Discussion
Benzo[α]pyrene to Benzo[a]pyrene (italic ”a”)
BαP to BaP
2.4. References and reagents
Benzo [a] anthracene to Benz[a]pyrene (italic ”a”)
benzo[k]fluoranthene to Benzo[k]fluoranthene (italic ”k”)
Benzo[b]fluoranthene to Benzo[b]fluoranthene (italic ”b”)
Dibenz[a,h]anthracene to Dibenz[a,h]anthracene (italic ”a,h”)
Benzo[ghi]perylene to Benzo[ghi]perylene (italic ”ghi”)
Indeno[1,2,3,cd]pyrene to Indeno[1,2,3-cd]pyrene (hyphen “-“ and italic ”cd”)
Define the compound of the internal standard compound, Philippine-d10
Philippine-d10 to Philippine-d10 (italic “d” and subscript “10”?)
2.6 Environmental risk and assessment approach
Equation (3): Sendiment to Sediment
3.2. Vertical distribution characteristics of PAHs
Define “LPAHs” and “HPAHs”.
Author Response
Dear Reviewer,
Thank you very much for reviewing our manuscript (ijerph-522967). We appreciate the reviewer’s positive comments on our work. We have carefully considered the comments from the Reviewer. We have revised the manuscript accordingly. The English language in the manuscript was polished and revised by LetPub, an author services company. In addition, we have carefully checked the entire manuscript to ensure that the formatting requirements meet the journal’s guidelines. Please find the revised manuscript as well as our responses (highlighted in red) to each of the Reviewer’s comments below.
Thank you for your time and consideration.
I wish you good health and much happiness.
Yours sincerely,
Huanling Wu
June 20, 2019
Overall Comments
Point: There are few studies that analyzes the relation with urbanization, although there are many papers on PAH distribution of sediment / soil, This work obtained several interesting results: the vertical PAH distribution was different depending on the use types; the environmental risk of PAHs was the highest in the ditch land; the PAH distribution depends on the build-up of urban areas, etc., which merit publication.
Response 1: We appreciate this Reviewer’s insight on our study.
Major comments
Point 1: Fig. 4 shows that the PAH distribution differed depending on the depth. Although the depth of sediment is an important parameter indicating the time, the relationship with the history of human activity has not been considered at all in Discussion.
Response 1: We thank this Reviewer for their helpful suggestion. As stated by the Reviewer, the sediment depth is an important parameter related to time. However, considering that the sediments in the lower reaches of the river may be disturbed by ships or dredging, we note that there may be interference with the original sedimentary sequence. In light of these potential errors, we did not determine a sedimentary time series.
Point 2: Also, the residue of PAH in sediments is considered to be different depending on the difference in the loading paths (inflow/falling), presence or absence of artificial work, decomposition reactions (microbe/photolysis) etc.However, Fig. 4 only describes that the type and concentration of PAH are different depending on the depth, and the relationship with differences in load paths, artificial work, decomposition reactions etc. is not considered in Discussion.
Response 2: We are interested in this Reviewer’ comments and suggestions. Based on the existing literature, only some internal and external factors are typically considered. Therefore, in this study, we included relationships between PAHs and TOC in sediments, sediment depth, land use type and distance from the built-up area. Effects of artificial work and decomposition reactions, among other factors, on PAHs will be investigated in our future work.
Minor comments
Point 1: Abstract, Fig. 4 caption and 4. Discussion
Benzo[α]pyrene to Benzo[a]pyrene (italic ”a”);BαP to BaP
Response 1: This has been corrected. Please see page 10, line 261; page 12, lines 313, 314, and 317; and page 13, line 345.
Point 2: 2.4. References and reagents
Benzo [a] anthracene to Benz[a]pyrene (italic ”a”)
benzo[k]fluoranthene to Benzo[k]fluoranthene (italic ”k”)
Benzo[b]fluoranthene to Benzo[b]fluoranthene (italic ”b”)
Dibenz[a,h]anthracene to Dibenz[a,h]anthracene (italic ”a,h”)
Benzo[ghi]perylene to Benzo[ghi]perylene (italic ”ghi”)
Indeno[1,2,3,cd]pyrene to Indeno[1,2,3-cd]pyrene (hyphen “-“ and italic ”cd”)
Philippine-d10 to Philippine-d10 (italic “d” and subscript “10”?)
Response 2: Corrected. Please see page 5, lines 144–146 and 151.
Point 3: Define the compound of the internal standard compound, Philippine-d10
Response 3: We have added the definition of Philippine-d10. Please see page 5, lines 152 and 153.
Point 4: 2.6 Environmental risk and assessment approach
Equation (3): Sendiment to Sediment
Response 4: Revised. Please see page 5, Equation (3).
Point 5: Define “LPAHs” and “HPAHs”.
Response 5: Thank you for your suggestion. Please see the definitions on page 8, lines 235–237.
Round 2
Reviewer 1 Report
Authors revised manuscript according to the reviewer's comments. Manuscript in its present form is now suitable for publication.
Reviewer 3 Report
The author have condidered carefully the comments given by the reviewers. The revised manuscript is acceptable for publication.